# Pathogenesis-Related Proteins (PRs) with Enzyme Activity Activating Plant Defense Responses

**DOI:** 10.3390/plants12112226

**Published:** 2023-06-05

**Authors:** Cristiane dos Santos, Octávio Luiz Franco

**Affiliations:** 1S-Inova Biotech, Pós-Graduação em Biotecnologia, Universidade Católica Dom Bosco, Campo Grande 79117-900, Brazil; ocfranco@gmail.com; 2Centro de Análises Proteômicas e Bioquímicas, Pós-Graduação em Ciências Genômicas e Biotecnologia, Universidade Católica de Brasília, Brasília 71966-700, Brazil

**Keywords:** plant–pathogen interaction, plant protection, preformed mechanism, postformed mechanism, signaling pathways

## Abstract

Throughout evolution, plants have developed a highly complex defense system against different threats, including phytopathogens. Plant defense depends on constitutive and induced factors combined as defense mechanisms. These mechanisms involve a complex signaling network linking structural and biochemical defense. Antimicrobial and pathogenesis-related (PR) proteins are examples of this mechanism, which can accumulate extra- and intracellular space after infection. However, despite their name, some PR proteins are present at low levels even in healthy plant tissues. When they face a pathogen, these PRs can increase in abundance, acting as the first line of plant defense. Thus, PRs play a key role in early defense events, which can reduce the damage and mortality caused by pathogens. In this context, the present review will discuss defense response proteins, which have been identified as PRs, with enzymatic action, including constitutive enzymes, β-1,3 glucanase, chitinase, peroxidase and ribonucleases. From the technological perspective, we discuss the advances of the last decade applied to the study of these enzymes, which are important in the early events of higher plant defense against phytopathogens.

## 1. Introduction

As is well-known, higher plants are sessile organisms, and this condition could be a weak point in their defense against stresses. Nevertheless, over the course of evolution, a highly specialized defense system developed [1]. The plant defense mechanism is composed of a complex system that amplifies chemical and molecular signals [2,3]. It is known that the first line of plant defense against pathogen attack can involve immediate immune system responses with pattern-triggered immunity (PTI). When this first reaction is ineffective, other factors may be triggered by the effector-triggered immunity (ETI), which usually provides plant resistance (Figure 1). In ETI, plants are directly or indirectly stimulated by effectors from pathogens, resistance proteins that will trigger a more effective defense response, quickly, through a sophisticated defense network [4,5].

This complex defense network consists of combined mechanisms, preformed (constitutive) and postformed (induced) mechanisms, that are capable of protecting plants against several stresses including pathogen attack [2,6,7,8]. Among these post-formed and performed mechanisms, proteins that play an essential role in plant defense are found. Briefly, in the initial infection stages, plants produce proteins that accumulate, and many of these proteins have antimicrobial activity and can induce the production of structural compounds, including lignin and callose deposition [7]. Such proteins can act in the hypersensitivity reaction, accelerating cell death, to limit the propagation of phytopathogens. Among these proteins, the pathogenesis-related (PR) proteins stand out (Table 1), which are capable of inducing the plant’s innate immune system [4,5,9,10].

These proteins can have direct or indirect action in plant resistance against microorganisms. PR proteins can inhibit pathogen growth and/or spore germination, and can also act as antimicrobial agents, hydrolases, and proteinase inhibitors and perform other activities [45,46]. PR defense proteins are molecules with different molecular weights, ranging from 6 to 43 kDa. They are thermostable, soluble at pH < 3, and protease-resistant, thus contributing to quantitative changes in protein levels during defense responses [9,18,25]. It is believed that PR proteins are encoded, but it is only after the presence of a stimulus that they will be expressed in plants. The stimuli that can lead to the expression of such proteins include infection by pathogens, the induction of resistance by elicitors, the accumulation of high plant hormone concentrations and some stresses, including cytoplasm disruption [10,47].

In general, elicitor molecules are found naturally in the cell wall of pathogens (Figure 1), and some PR proteins can hydrolyze these polysaccharides. The action of these proteins transforms these polymers into eliciting oligosaccharides. Thus, elicitors can induce three types of resistance, including local acquired resistance (LAR), acquired systemic resistance (ASR) and induced systemic resistance (ISR). The latter can be considered the most important [10,47,48,49].

In ASR plant defense, induction can occur through the modification of the cell wall and phytoalexin production, besides inducing the expression of several plant defense genes involved, including the expression of PR-producing genes. PR proteins can prevent pathogen colonization in plant cells, containing infection by efficiently activating host defense mechanisms [50]. Among the PRs, some have enzymatic activity, such as β-glucanases (PR-2), chitinases (PR-3, PR-8, PR-11), peroxidases (PR-9), and ribonucleases (PR-10) [10,26,40,47]. These enzymes and their participation in plant protection against pathogenic agents are highlighted below.

Currently, about 19 families of PRs have been reported (Table 1), including β-1,3-glucanases, chitinases, thaumatin-like proteins, peroxidases, ribosome-inactivating proteins, defensins, nonspecific lipid transfer proteins, oxalate oxidase, and oxalate-oxidase-like proteins, amongst others [46,51]. In this context, the present review will discuss PRs with enzymatic activity (β-1,3-glucanase, chitinase, peroxidase and ribonuclease), important in the early events of higher plant defense against phytopathogens, and also the technological advances of the last decade, applied to the study of these proteins. Technological innovations have allowed the study of natural enzymes, mainly with advances in recombinant DNA technology and protein engineering. These tools have been important for enzyme studies, especially for understanding sequence–function relationships. In addition to being useful for prospecting and producing natural enzymes, biotechnological tools contribute to the obtention of bioinspired enzymes. These enzymes can be obtained with improved characteristics for application in agriculture, including the production of plants with improved characteristics to tolerate different stresses. In addition, these tools can be useful in enzyme production for more sustainable applications in industrial processes and bioremediation methods [52,53,54].

## 2. PR Proteins with Enzymatic Action and the Role in Plant Defense Activation

### 2.1. PR-2 and PR-3 Families: β-1,3 Glucanases and Chitinases

β-1,3 glucanases belong to PR-2 family, classified as endonuclease enzymes (E.C.3.2.1.39). They are multifunctional enzymes present in many living beings, including bacteria, fungi and some invertebrate animals and plants. Over the years, four β-glucanases subfamilies have been reported (A, B, C and D). Among these subfamilies, ten β-1,3-glucanases were classified, based on amino acid sequences shared considering similarities and uniqueness [50]. The β-1,3 glucanase enzyme is one of three β-glucans found in plants (in addition to β-1,4 glucanases and β-1,3-1,4 glucanases) [55].

Despite being distributed differently among plant organs, glucanases (β-1,3) may play an important role in the physiological systems of plants, including plant growth, seed germination re-production, and fruit ripening [56,57,58]. β-glucans (cellulose, callose, xyloglucan, vmixed-linked glucan—MLG) are cell wall structures, predominant in almost all vegetables. These structures can be degraded by specific enzymes such as β-glucanases [59]. Due to their great potential in plant defense participation, β-1,3 glucanases have been extensively studied, isolated and sequenced [10,47,60,61]. Naturally, β-1,3 glucanase gene expression levels are relatively low, but when a plant–pathogen interaction or elicitors are used, high levels of β-1,3 glucanase can be detected; enzyme accumulation occurs rapidly and consequently hydrolytic activity increases [10,47].

In a fungal invasion, for example, specifically in fungal cell wall degradation through the action of β-1,3 glucanases (Figure 1), oligomers are released, namely β1,3/1,6-D-glucan. These released oligomers can be considered elicitor oligosaccharides. The release of these elicitors induces a plant defense response, demonstrating direct antimicrobial activity [10,47]. Using DNA recombinant technology, a novel β-1,3-glucanase (Gns6) was characterized by functionality. β-1,3-glucanase Gns6 belongs to subfamily A. The gene expression of Gns6 was evaluated at an early stage of rice blast infection, and the involvement of β-1,3-glucanase Gns6 in early plant defense was proved [50]. In order to evaluate the effect of β-1,3-glucanase on the construction of the fungal cell wall, the phytopathogenic fungus was submitted to Gns6 in an antifungal activity bioassay. The results revealed that the enzyme Gns6 exhibited potent antifungal activity against *Maganaporthe orzyae*, which causes blast disease in rice [50].

Plant β-1,3 glucanases can act in synergism with chitinases, catalyzing the cell wall degradation of microorganisms through the process of hydrolysis of β-1,3 glucans and chitin, respectively (Figure 1). These enzymes are the most studied among the PRs. Furthermore, these enzymes, together with other hydrolases, also participate in the degradation of cell membrane constituents, mainly fungi [59]. Chitinases are enzymes (E.C. 3.2.1.14) belonging to groups 3, 4, 8, and 11 of the PRs.

Chitinases can also act similarly to chitosanases (induced in plants as a response to pathogenic interaction) and are capable of degrading chitosan, which is present in structural components of the cell wall of some species of fungi, including those of the order Mucorales. Chitinases have efficient action in the degradation of chitin, the second-most abundant structural polysaccharide in nature, found in insect exoskeletons; they are also vital components of the fungal cell wall. Additionally, chitinases can be observed in some plant species in response to the action of phytopathogenic viruses [62].

Some chitinases identified so far have demonstrated lysozyme activity, which may also act in bacterial cell wall degradation. This may be antibacterial action, demonstrated by the ability to hydrolyze the β-1,4 bonds that are between N-acetylmuramic acid and N-acetylglucosamine in peptidoglycan-like heterosaccharides present in the cell wall of prokaryotes [63]. As mentioned, during pathogen–plant interactions, elicitor molecules are recognized, and just like β-1,3-glucanase, the chitinases can hydrolyze these elicitors, transforming them into eliciting oligosaccharides (Figure 1). So, the elicitors produced from chitin and β-1,3-glucanase activity can activate a signaling network, where defense genes are activated to produce other PR proteins that accumulate and act in pathogen cell degradation [10,47,48,49].

Functional analysis has revealed that transgenic plants of *Arabidopsis*, overexpressing the endochitinase gene, proved to be resistant to *Xanthomonas campestris* pv. *campestris* (*Xcc*) when compared to wild-type plants. This endochitinase was identified in cabbage plants and showed up-regulation 24 h after infection *Xcc*. Gene expression analysis showed high levels of the endochitinase gene when compared to the uninoculated cabbage plant [21]. The analysis of *Cucumis sativus* L. showed the induction of genes encoding chitinase in plant roots during infection by *Fusarium oxysporum* f. sp. cucumerinum (Foc) [64].

An in vitro assay with purified chitinases Chi2 and Chi14 showed that proteins limited Foc growth. In addition, the gene silencing of Chi14, using the technique of virus-induced gene silencing (VIGS), increased the plant’s sensitivity to fungus. Chi2 gene silencing drastically compromised the activation of the jasmonic acid pathway gene, which is a phytohormone important in plant defense signaling. These results corroborate the hypothesis that chitinase (Chi2) may play a key role in plant resistance [64]. The overexpression of type II chitinase (LcCHI2) in *Leymus chinensis* conferred increased hydrolytic activity in transgenic tobacco and corn plants, which have been shown to be more resistant to pathogens and salt stress [65].

Twenty-six chitinase genes were identified in *Morus notablis* plants [66]. The differential expression of one of these enzymes, MnChi18, leads to an increased defense against *Botrytis cinera.* The plant models overexpressing MnChi18 were protected from damage and were shown to be involved in *B. cinera* resistance [66]. Finally, another study showed that the chitinase gene can positively regulate the hypersensitive and defense responses of *Capsicum annuum* L. to infection caused by *Colletotrichum acutatum* [67].

In summary, both β-1,3-glucanase and chitinase have been shown to play a significant role in plant defense against microbial agents. It is known that β-1,3-glucanases accumulate during pathogen attack and can act in the hydrolysis of the pathogen cell wall. As mentioned above, the substrate for this enzyme, β-1,3-glucans, can be found in several microorganisms [50]. Faced with the action of β-1,3-glucanases and chitinase enzymes, oligomers are released, which are β1,3/1,6-D-glucano and chitin, respectively. These released oligomers can be called elicitor oligosaccharides. Elicitor release induces a plant defense response. The activity of both enzymes can cause the depolymerization of structural saccharides present in the pathogen wall, degrading it [68].

### 2.2. PR-9 Family: Peroxidases

Together, plant peroxidases, β-1,3-glucanases and chitinases act in the early plant infection stages [69]. Once the plant has detected pathogen elicitors or abiotic stress, a series of events, such as oxidative burst, takes place in an attempt to protect the plant from damage induced by ROS. Plant ROS production leads to oxidative burst (Figure 1). This action plays an important role in direct defense by promoting lignification and pathogen intoxication due to ROS accumulation [69,70,71].

Currently, the peroxidases are classified into two groups, the first of which is nonheme peroxidase, which is found in prokaryotes and eukaryotes (including halo-peroxidases, NADH peroxidases, thiol peroxidases, and alkylhydro-peroxidase). The second group, heme-peroxidases, is composed of two superfamilies: (1) the peroxidase-cyclooxygenase superfamily (PCOXS) and (2) the peroxidase-catalase superfamily (PCATS). The PCOXS representatives are known as the animal-peroxidase superfamily, while the PCATS are commonly called the nonanimal heme peroxidases [72,73]. Nowadays, three classes are found in the nonanimal peroxidases: class I (ascorbate peroxidase, yeast cytochrome and bacterial catalase peroxidases), class II (heme peroxidase, includes lignin peroxidase, manganese peroxidase and versatile peroxidase) and class III (found in plants). The class III peroxidases correspond to about 70% of plant-derived peroxidases [73]. Plant peroxidases (POX, EC 1.11.1.7) are antioxidant enzymes, belonging to group 9 of the PRs (PR-9) [40].

Plant peroxidases have an important role in plant physiology (Figure 1), including lignification and wound healing; these enzymes can also participate in the regulation of cell elongation [74]. Peroxidases play a plant defense role against pathogens. Besides participating in cell signaling after infection, peroxidases can polymerize macromolecules which, after being deposited on the extracellular surface, can promote cell wall strengthening and thus make pathogen invasion more difficult. Peroxidases can also induce the oxidative degradation of phenolic compounds in the cell rupture region caused by pathogens in the first infection stages [74]. Along with two other oxidizing enzyme families (unrelated), the laccases (LACs) and the polyphenol oxidases (PPOs) family, the PRXs make up the phenoloxidases. PRXs can oxidize substrates, including some phenols, through the reduction of H_2_O_2_ or organic peroxides [74,75,76]. Some phenols can generate oxygen radicals, which can be extremely reactive and harmful to the plant. PRXs and other phenoloxidases play a protective role, leading to the oxidative degradation of some phenol forms at the site of infection [76]. Thus, the use of plant peroxidases arouses great industrial interest as potential biodegradable agents. PRX from plants can degrade residual phenols in water, from industrial wastewater [77,78,79].

Diverse isoforms of the peroxidase family are found throughout the plant and are capable of oxidizing numerous molecules. POXs are involved in many biological activities (Figure 1), such as cellular detoxification and the elimination of ROS (including ^1^ O_2_, singlet oxygen; O_2_^•−^, superoxide anion; H_2_O_2_, hydrogen peroxide; and OH^•^, hydroxyl radical). Peroxidases are important enzymes for maintaining the redox homeostasis of plant cells [29].

In addition, in the first moment of plant stress, ROS are produced to protect the plant from oxidative stress, since the oxidative burst can be lethal to the plant. The balance between antioxidant (AOX) and ROS production is necessary for plant normality. Plant detoxification is also needed, and this can be through peroxidase and other enzymes, such as catalase, superoxide dismutase, etc. [80]. An imbalance between AOX and ROS, either due to AOX depletion or ROS excess, can prolong oxidative stress, which can compromise the production of lipids, amino acids, proteins, nucleotide acid and pigments [81]. The remaining oxidative stress also causes cellular damage, leading to membrane injury, organelle function losses, reduced metabolic efficiency, reduced carbon fixation, electrolyte leakage, and chromatid breaks and mutation. All this damage can lead to growth reduction, yield loss and cell death. Peroxidase action is essential for maintaining cellular balance [81].

Both peroxidase and NADPH oxidase, present in the plant cell wall, play an important role in the apoplastic oxidative burst after microbial attack against plants. In *Arabidopsis*, after interaction with the fungus *Alternaria brassicola*, cell wall peroxidases (named PRX33 and PRX34) and NADPH oxidase mediated the oxidative burst in the plant. These enzymes are considered to be the main catalyst of the oxidative burst process [82]. A characterization study also demonstrated that the *Arabidopsis* mutant prx34 can reduce ROS and callose accumulation after Flg22-elicitor treatment. These results corroborate other findings, showing that the PRX34 enzyme could be an important component for plant disease resistance [83].

Another work overexpressed a peroxidase (swpa4) gene, and the stress-related functions of these enzymes in *Ipomoea batatas* L. were evaluated. The results indicated that swapa4 gene overexpression can protect the plant from damage [28]. Furthermore, these results suggested that transgenic sweet potato, overexpressing the PRX genes, can respond more efficiently to saline stress [28]. An in vivo bioassay evaluation, using transgenic *Arabidopsis*, showed that lines overexpressing PRX genes (cotton gene *Gh*PRXIIB) were capable of tolerating and limiting nematode infection [84].

Some POX enzymes, such as ascorbate peroxidase (APX) and glutathione peroxidase (GPx), can catalyze the conversion of H_2_O_2_ to H_2_O [85,86]. Many APX isoforms can be found in different subcellular compartments, including chloroplasts, mitochondria, peroxisome and cytosol [87], and can play an important role in oxidative defense metabolism [88,89]. Transgenic plants overexpressing glutathione peroxidase (named AtGPXL5) revealed that this enzyme gene can participate in ethylene (ET) biosynthesis and signaling [90].

After treatment with the ET-precursor (1-aminocyclopropane-1-carboxylic acid—ACC), transgenic plants show glutathione- and thioredoxin-induced activity and other enzymes involved in ROS processing, which suggests the involvement of the AtGPXL5 gene with ethylene signaling and thus also with plant cell defense [90]. Transgenic plants of *Citrus sinensis*, overexpressing the CsPrx25 gene and encoding a class III peroxidase, show ROS homeostasis and increased H_2_O_2_ levels and consequently a strong hypersensitivity reaction to *Xcc*. The results also show that CsPrx25 gene overexpression contributes to the lignification process of the cell wall, increasing plant resistance [91].

In summary, ROS production increases, such as H_2_O_2_, and seems to protect the plant against environmental stimulus including pathogens, but can cause significant stress, since ROS accumulation can lead to cellular toxicity. APX and GPx can act in cellular homeostasis under oxidative stress, protecting the plant [86,92].

### 2.3. PR-10 Family: Ribonucleases

As mentioned, the ribonucleases (RNase) are a member of group 10 of PRs. They show approximately 17 kDa and exhibit a hydrophobic core capable of binding a wide variety of ligands. Ribonuclease PR10 has demonstrated ligand ability for low-molecular-mass compounds. The PR-10 hydrophobic cavity can bind with small molecules, and this hydrophobic cavity can be considered as a general feature of such enzymes [93]. Many studies have reported the protein–ligand interaction of PR-10 proteins, as reviewed by [93]. These enzymes can bind to steroids, cytokinin, flavonoids and fatty acids, phytoprostanes, phytomelatonin, gibberellic acid and plant metabolites, with molecules involved in flavor production and color. Ribonuclease enzymes can interact with phytohormones in the hormone-mediated signaling process [93,94].

Homologs share this conserved structure, but function is not a universal characteristic among the members of the group. These enzymes have been identified in different plant species. However, no unique biological function has been assigned to PR-10 proteins. Among the functions assigned to PR-10 are plant growth and development, as well as antioxidation, UV protection, and pathogen defense. An unusual protein was found in rubber trees and presented activity like the PR-10 class. After structural characterization, plant protection against the *Rigidoporus microporus* fungus was related to this protein. The structural analysis demonstrated that these proteins can bind with a deoxycholic acid ligand [95]. Deoxycholic acid is a bile acid (bile acid deoxycholic acid—DCA), which demonstrated action related to plant defense response. DCA can induce defense in *Arabidopsis* plants and reduce bacterial proliferation [96]. It is possible to observe the up-regulation of PR10 enzymes during plant pathogen interaction and/or direct induction after applying external phytohormones, proving the protective action of PR10 during plant–pathogen interaction [94,97,98]. Additionally, it is possible to observe an increase in PR10 abundance during interactions caused by viruses and fungi [45,99,100,101,102].

Besides PR-10′s involvement in the signaling pathways of defense genes, ribonucleolytic activity to cleave invading pathogens has been reported (Figure 1), causing the pathogen’s RNA cleavage [103]. During pathogen infection, the RNase activity of PR10 proteins can cause a cytotoxic cell impact and inhibit pathogen growth, degrading the pathogen cell [45,99,100,101,102]. This inhibition occurs mainly through ribonuclease penetration into the pathogen, with PR10 phosphorylation subsequently occurring, and consequently the destruction of pathogenic cell RNAs [25].

RNase activity can be exhibited by several PR-10 proteins but it is not believed to be a universal characteristic [104]. RNase activity is required under biotic and abiotic stress, since these proteins are involved in plant HR signalization, in programmed cell death control and/or apoptosis process [105,106]. Much evidence has been reported on the general activity of PR-10 against different phytopathogens such as fungi, bacteria, and viruses [93,103]. Additionally, one report indicated the protease inhibitory activity of PR-10 in the root-knot nematode *Meloidogyne incognita* [107].

Concerning the PR-10 activity against pathogens, although not well-explained, these enzymes are believed to be related to the endogenous cytokinin (CK) concentrations and CK in negative feedback regulation. These cytokines are involved in plant immunity modulation, acting directly in the plant defense response to many pathogens [98,103,108].

In addition, PR-10 proteins can interact with plant hormones such as ABA, JA, auxins, ethylene, and SA, which are involved in hormone-mediated signaling to mitigate damage suffered by the plant, caused by biotic and abiotic stress [103,109]. In plants infected with *Verticillium dahlia*, PR10 genes were found to be up-regulated after an expression profile investigation in leaves, roots and stems of strawberry plants [97]. Once again, the induction of some phytohormones, including ABA, SA, JA, and gibberellic acid, was seen in the early stages of plant–pathogen interaction. In roots, just two hormones were induced, indole acetic acid (IAA) and JA, but in the late stages of infection [97].

In *Valsa mali* fungus, VmP1 is a virulence factor and can interact with PR10 (named MdPR10) in vivo. The MdPR10 gene present in *Malus domestica* is a VmP1 target, and when the silencing of the MdPR10 gene occurs, plant susceptibility increases in plants, while gene overexpression decreases the infection, showing the role of MdPR10 in plant defense against *V. mali* [110]. Mahmoud et al. (2020) [31] demonstrated, with exogenous products to induce the plant’s systemic resistance, that after treatment the plants showed increased abundance of PR10 as well as another protein, which is also used as a marker of systemic acquired resistance (SAR), namely, phenylalanine ammonia lyase (PAL). The in vivo results demonstrate antiviral activity against the tomato pathogen Tobacco Mosaic Virus (TMV) [31].

## 3. Concluding Remarks

During plant–pathogen interaction, a sophisticated signaling network results in gene induction to produce several molecules for plant defense. Among these molecules, several protein types are produced, including pathogen-related enzymes, such as glucanases, chitinases, peroxidases and ribonuclease [25]. These enzymes can be produced in trace amounts by healthy plants, but they may have increased levels in the face of a pathogen. These enzymes can act directly or indirectly in plant defense, leading to phytopathogen death or the induction of other defense response routes [10,47,49,103,111]. The use of these enzymes has been extensively studied to determine resistance induction strategies in plants. The main biotechnological advances in plants have been promising and have elucidated resistance mechanisms. These advances have showed enlightening findings regarding this enzyme’s integration into the plant defense signaling network. Improvements in biotechnological techniques ensure the transformation of transgenic plants and promote the isolation of many genes induced in pathogenesis. These improvements allow the genetic manipulation of plants to exhibit resistance to a broad spectrum of pathogens. Genetic engineering techniques can manipulate plant genes and further evaluate the effects singly or synergistically, against several pathogens, resulting in engineering disease-resistant plants.

## Figures and Tables

**Figure 1 plants-12-02226-f001:**
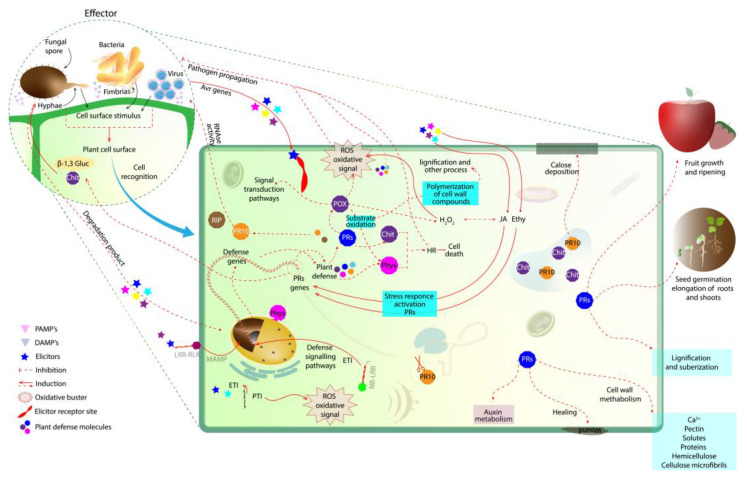
Initial plant defense response in plant–pathogen interaction. At first, the pathogen is recognized on the plant cell wall surface. Then, the elicitors activate a signaling network, where defense genes are activated to produce PR proteins that accumulate and act in the degradation of the pathogenic cell (i.e., β-1,3-gluc and chit; the degradation products of these enzymes can also act as elicitors). The oxidative buster mediates the generation of ROS in an attempt to limit the spread of the pathogen. PRs (i.e., β-1,3-glucanases, chitinases, PRXs, PR10 together with Phyx) are able to induce a hypersensitivity response to prevent the spread of the pathogen to other tissues, releasing elicitors that induce the plant’s defense mechanism. PR10, RBPs and RIPs are also produced by the plant, mediating virus infection. NB-LRR and PR10 act together in gene defense induction. PRX, peroxidase; PRs, pathogen-related proteins; Chit, chitinase; ROS, reactive oxygen species; Phyx, phytoalexin; β-1,3-gluc, β-1,3-glucanases; Ethy, ethylene; JA, jasmonic acid; LRR-RLK, leucine-rich repeat receptor-like protein kinase; NB-LRR, nucleotide-binding and leucine-rich repeat; RBPs, RNA-binding proteins; RIPs, ribosome-inactivating proteins.

**Table 1 plants-12-02226-t001:** Pathogenesis-Related Proteins (sPRs) properties and roles in plant defense.

Family	Properties/Functions	References
PR-1 (11a, 1b and 1c)	Abundant proteins in the apoplast during plant–pathogen interaction Inhibit pathogens Antifungal and antivirus activity Metal tolerance	[11,12,13,14,15,16,17]
PR-2 (Classes: I, II, and III)	Plant cell wall (β-1,3-glucan hydrolysis) Antibacterial, antifungal and antivirus activity	[15,18,19,20]
PR-3; PR-4; PR-8; PR-11 (Classes: I, II, IV, V, VI, and VII)	Plant cell wall (Chitin hydrolysis) Antibacterial and antifungal activity Salt tolerance	[15,18,20,21]
PR-5	Similarities with thaumatin Antifungal activity Cause osmotic rupture of fungal plasma membrane	[18,22,23]
PR-6	Protease inhibitors Cleave exopeptidases produced by bacteria, fungus and insects	[24,25,26]
PR-7	Endoproteases Mechanism of action is understudied Might be antimicrobial (pathogen cell wall degradation)	[25,26,27]
PR-9	Peroxidase activity Catalyze the oxidation of hydrogen peroxide on substrates (organic and inorganic)	[28,29]
PR-10	Ribonucleases—degrade RNA Programmed cell death during hypersensitivity reaction Antibacterial, antifungal, antinematode and antivirus activity Slat and cold stress tolerance	[20,30,31,32]
PR-12	Defensins Produced constitutively in plant structures (leaves, flowers, tubers, pods and seeds) Increased abundance during plant–pathogen interaction Antibacterial activity	[15,18,26]
PR-13 (Classes: I, II, III and IV)	Thionins—bacterial membrane lysis Distributed in the plant cell wall, vacuole and protein bodies Defense against a wide range of pathogens	[26,33,34,35,36]
PR-14	Nonspecific lipid transfer proteins (ns-LTPs)—cuticle synthesis Secreted and are associated with plant cell wall Defense against a wide range of pathogens	[26,37,38,39]
PR-15; PR-16	Oxalate oxidase and oxalate-oxidase-like protein (OLP’s), respectively Generation of ROS immediately after pathogen attack, which has antimicrobial activity	[25,26,40,41]
PR-17	Similarities with aminopeptidase (such as that found in eukaryotes and bacteria) Secretory protein Proteolytic activity Antifungal and antiviral activity	[18,25,42]
PR-18	Carbohydrate oxidases properties Substrate specificity resulting in hydrogen peroxide as one of the reaction products Antibacterial activity	[25,43]
PR-19	Biological role is not deciphered yet Antimicrobial activity	[25,44]

## Data Availability

No new data were created or analyzed in this study. Data sharing is not applicable to this article.

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
