# Peer review of "Pathogenesis-Related Proteins (PRs) with Enzyme Activity Activating Plant Defense Responses"

_plants, 2023, doi:10.3390/plants12112226_

Round 1

Reviewer 1 Report

The review deals with pathogenesis-related proteins with enzyme activity in early events activating plant defense responses.

In my opinion the introduction  and concluding remarks are well organised, whereas paragraph 2 could be improved to better explain the topics. In particular, I suggest inserting a table with the proteins described in the manuscript to summarize their characteristics and properties. This could help to get an overview and summarize the informations making it easier to read.

On the contrary, the table that the authors have included in the manuscript is not necessary since it concerns general defense mechanisms which are not the topic of the review. Anyhow, there is a mistake in this table since it is indicated as Table2 instead of Table 1 (see line 40) and it is not clear why postformed mechanisms are not indicated.

In addition, the authors speak of technological perspectives (line 20) and technological advances (line 104) but it is not clear what they refer to, they must explain the concept.

Line 98: The authors cite table 1, is that right? Table 1 concerns preformed mechanisms not PRs’ family.

Line 106: Add in title "in the early events of plant defence”.

Paragraph 2.1 : I suggest to better organize the text. The concept that glucanases degrade beta-glucans is repeated at several points, eliminates repetition. In addition, I suggest moving above lines 131-140.

Paragraph 2.2: Lines 196-197 are redundant.

                          Lines 222-223: the authors should better explain the effect of degradation of phenolic compounds.         

                          Line 226: the cited figure 3 is missing and the figure 2 does not exist.

                          Line 229: what kind of equilibrium?

Author Response

We thank you for the careful reading of your text and we are grateful for your consideration of this manuscript. We also very much appreciate your suggestions, which have been very helpful in improving the manuscript. In any case, we are open to consideration of any further comments on our answers. All modifications were answered point by point and listed above. Please see the attachment.

Reviewer 2 Report

The review paper entitled: Pathogenesis-related proteins (PRs) with enzyme activity in early events activating plant defense responses,  by the authors Dos Santos and Luiz Franco aims to summarise the role of PR proteins during early plant defense activation. 

General comments: The title of the above-mentioned review suggests the authors will discuss deeply the role of PR proteins "in early events activating plant defense". I consider as early events activating plant defense the intracellular accumulation of calcium, the formation of PRR complexes and the protein phosphorylation mediated by MPKs, however I think the manuscript did not cover this aspect. I, therefore, suggest the authors to modify the title by removing “early events”. Moreover, over the last decades, one of the most studied and referred PR protein family is the PR1 class.  On the other hand, throughout the entire manuscript, I have not found any comments about this important family. Was it made on purpose? Can the authors explain this point? 

Specific comments and suggestions about the manuscript:

It is generally accepted that the plant immune system is made of, at least, two defensive layers defined as Pattern Triggered Immunity (PTI) and Effector Triggered Immunity (ETI), which somehow is also indicated in Figure 1. Please describe these lines of defense briefly in the Introduction putting into context also the PRs activation in this scenario.

Line 40: Table 2. Please correct it in Table 1 and formulate the proper title. 

Figure 1: the schematic representation is too messy, please, simplify it (for example I think it's not necessary to represent the MAP kinase pathway) and increase the size of the main actors described in the manuscript. Note that callus is a growing mass of unorganized plant cells, whereas callose is the polysaccharide reinforcing the cell wall. Please correct this slip.

Please retitle all the result sections emphasizing first the class of PR which will be described. For example in section 2.1. PR2 and PR3 families: the role of B-1,3 glucanases and chitinases in plant defense activation. Same for the other titles.

Line 141: please start introducing the 1,3 glucanases as belonging to the PR3 (or others) families

Line 264: please report the full name of the Ethylene precursor used in the cited study. I guess it was ACC.

From Line 286, please also cite and discuss briefly the recent papers published by Longsaward et al., 2023, BMC plant Biology, in which they discussed the possibility of PR10 to bind the steroid deoxycholate, a plant defense elicitor molecule as reported by Zarattini et al 2017 Mol plant path.

To help the readers, please replace reference 52 with an English-written article

Author Response

(The authors gave the same response as above.)

Round 2

Reviewer 1 Report

The review deals with pathogenesis-related proteins with enzyme activity in early events activating plant defense responses.

In my opinion the revised manuscript can be accepted in present form.

Reviewer 2 Report

The current version of the review deserves to be publish. Congrats to the authors